

# Disaggregating a Regional Extent Digital Soil Map using Bayesian Area-to-Point Regression Kriging for Farm-Scale Soil Carbon Assessment

Sanjeewani Nimalka Somarathna Pallegedara Dewage [1,2], Budiman Minasny[1], Brendan Malone [1,2],

1 Sydney Institute of Agriculture, School of Life and Environmental Sciences, The University of Sydney, 1 Central Avenue, Australian Technology Park, Eveleigh, NSW, 2015, Australia.
2 CSIRO Agriculture and Food, Australia.

*Correspondence to*: Sanjeewani Nimalka Somarathna Pallegedara Dewage (sanjeewanis@gmail.com)

**Abstract**

Most soil management activities are implemented at farm scale, yet digital soil maps are commonly available at regional/ national scale. Disaggregating these regional/national maps to be applicable for farm scale tasks particularly in data poor or limited situations. Although disaggregation is a frequently discussed topic in recent DSM literature, the uncertainty of the disaggregation process is not often discussed. Underestimation of inferential or predictive uncertainty in statistical modelling leads to inaccurate statistical summaries and overconfident decisions. The use of Bayesian inference allows for quantifying the uncertainty associated with the disaggregation process. In this study, a framework of Bayesian Area-to-Point Regression Kriging (ATPRK) is proposed for downscaling soil attributes, in particular, maps of soil organic carbon. Estimation of point support variogram from block supported data, was carried out using the Monte Carlo integration via the Metropolis-Hasting Algorithm. A regional soil carbon map with a resolution of 100 m (block support) was disaggregated to 10 m (point support) information for a farm in the northern NSW, Australia. The derived point support variogram has a higher partial sill and nugget while the range and parameters do not deviate much from the block support data. The disaggregated fine-scale map (point support with a grid spacing of 10m) using Bayesian ATPRK had an 87% concordance correlation with the original coarse scale map. The uncertainty estimates of the disaggregation process were given by a 95% confidence interval (CI) limits. Narrow CI limits indicate, the disaggregation process gives a fair approximation of mean SOC content of the study site. The Bayesian ATPRK approach was compared with dissever; a regression-based disaggregation algorithm. The disaggregated maps generated by dissever had 96% concordance correlation with the coarse-scale map. dissever achieves this higher concordance correlation through an iteration process while Bayesian ATPRK is a one-step process. The two disaggregated products were validated with 127 independent topsoil carbon observations. The validation concordance correlation coefficient for Bayesian ATPRK disaggregation was 23% while downscaled maps generated from dissever had 18% CCC. The advantages and limitations of both disaggregation algorithms are discussed.



# 1 Introduction

The proliferation of digital soil mapping (DSM) and modelling techniques during the past decade has made available vast amounts of digital soil information (Lagacherie, 2008; Grunwald, 2009; Minasny and McBratney, 2016). For example, the GlobalSoilMap project (Arrouays et al., 2014) aspired towards delivering a global coverage of soil attributes at 100 m resolution for all five standard soil depths. The Soil and Landscape grid of Australia (Grundy et al., 2015), maps of topsoil organic carbon content across Europe (de Brogniez et al., 2015), national digital soil maps of France (Mulder et al., 2016), and carbon map of Nigeria (Akpa et al., 2016) are examples of large extent digital soil mapping (DSM). However, these national or regional digital maps do not capture the sufficient spatial variability of soil for farm-scale decision making (Malone et al., 2017). Yet these maps can be converted into useful farm scale information using a combination of fine-scale environmental covariates and geostatistical methods. This process is called downscaling or disaggregation (Malone et al., 2017). Disaggregation is the process of detailing information collected at a small or coarse scale towards a larger or finer scale (Cheng, 2008). Disaggregating techniques can be classified into to two categories; geostatistical and regression techniques. Area to point kriging (ATPK) (Kyriakidis, 2004) and its variants such as area to point regression kriging (ATPRK) are the most common geostatistical techniques applied to disaggregate soil resource information. Incorporating both point and areal data in the kriging system is such a variant of ATPK presented by Goovaerts (2011). That study demonstrated ATPK to map the variability within soil units and secondly introduced a general formulation for incorporating area-to-area, area-to-point, and point-to-point covariance in the kriging system. Replacing the areal mean with summary statistics in the kriging system is another improved ATPK technique suggested by Orton et al. (2014). Román Dobarco et al. (2016) extended this approach using summary statistics in the ATPRK system. Further, a Bayesian regression kriging approach was presented by Ramos et al. (2013). The model used a Gaussian process model, and it is a novel image fusion technique for panchromatic sharpening of low-resolution satellite images using images from unmanned aerial vehicle flown over agricultural fields.

Regression approaches are also a popular approach to downscale soil attributes. For example, Malone et al. (2012), formalised a statistical downscaling algorithm; dissever based on the Liu and Pu (2008) algorithm for downscaling continuous earth resources with a case study of downscaling soil organic carbon content. This method was refined by Roudier et al. (2017), allowing the user to choose between a range of regression models apart from the originally contained Generalised additive models (GAM). "DSMART" (Disaggregation and Harmonisations of Soil Map Units through Resampled Classification Trees) is another popular regression-based approach, proposed by Odgers et al. (2014) to downscale soil map units to soil series information. However it is different to the discussed method since there is no fine gridding involved.

Disaggregation is a change of support problem (Cressie, 1991; Gotway and Young, 2002). Spatial support is the geometrical size, shape orientation spatial unit of an observation or a prediction. In the disaggregation process, when the support changes the associated statistical and spatial properties of the variable also change, adding a major source of uncertainty (Truong et al., 2014). ATPK is a theoretically-based method that uses the classical kriging principles to predict at point support using block support information (Kyriakidis, 2004). Incorporating fine scale covariate data in the kriging system (ATPRK) is superior to ATPK as it allows for within class variability, spatial autocorrelation and also "scorpan" factors (McBratney et al., 2003) represented by ancillary covariates (Kerry et al., 2012).





In disaggregating soil information, quantifying the uncertainties of this process is not often discussed. Underestimation of inferential or predictive uncertainty in statistical modelling can lead to inaccurate statistical summaries and overconfident decisions (Draper, 1995). Therefore, it is important to account for the underling parameter uncertainty associated with the disaggregation process. The use of Bayesian inference in ATPRK system enables quantifying the uncertainty associated with inferring point support parameters from the block supported system, in addition to estimating the product uncertainty (Diggle et al., 1998).

To the best of our knowledge, Bayesian ATPRK has not been used for downscaling soil information. In this paper, we present a Bayesian ATPRK approach for downscaling soil attributes. Specifically, the study focuses on downscaling regional soil organic carbon maps to high-resolution farm scale maps. Soil organic carbon is arguably a key soil property due to its conferring benefits to the soil physical and chemical properties and potential to store atmospheric carbon. Accurate estimations of SOC stocks at farm scale are important for precision agricultural needs.  In addition, the fine-scale maps can be used for designing sampling strategies for quantification of SOC stocks (de Gruijter et al., 2016) required by carbon sequestration auditing process such as carbon farming initiatives in Australia  (Malone et al., 2017).

A regional soil carbon map with a resolution of 100 m (block support) was disaggregated into 10 m (point support) information for a farm in the Spring Ridge region in the northern NSW. Application of the Bayesian ATPRK approach is presented and demonstrated within the study. In addition, a comparative analysis with the dissever algorithm, a regression-based downscaling method, is performed to assess the performance of Bayesian ATPRK in the context of spatial downscaling of digital soil carbon mapping.

## 2 Theoretical background

Assuming we are observing a continuous Gaussian process, we denote the process as S(p) for locations $p \in D$, where D is the region of interest. For block support (B) data, where $B \subset D$, we presume that the observations are as block averages:

$$\mathbf{S}(B) = |B|^{-1} \int_B \mathbf{S(p)} \; d\mathbf{p} \tag{1}$$

where $|B|$ denotes the area of B.

The underlying Gaussian process S can be expressed as

$$\mathbf{S} = \sum_{k=0}^{l} \mathbf{m}\beta + \varepsilon \tag{2}$$

where $\mathbf{m}$ is the design matrix of the trend function and $\beta$ vector of coefficients, and $\varepsilon$ is the stochastic residuals with mean of zero and covariance function $C(\boldsymbol{\theta})$, where $\boldsymbol{\theta}$ represent the parameters of the covariance function. Since the block supported observations; $\mathbf{S_B}$ and point support observations; $\mathbf{S_p}$ are jointly Gaussian, the prediction problem can be introduced as:

$$\begin{bmatrix} \mathbf{S_p} \\ \mathbf{S_B} \end{bmatrix} \sim N \left( \begin{bmatrix} \mathbf{m} \\ \mathbf{M} \end{bmatrix} \beta, \begin{bmatrix} \mathbf{C_{pp}} & \mathbf{C_{pB}} \\ C_{Bp} & C_{BB} \end{bmatrix} \right) \tag{3}$$





where $\mathbf{C_{pp}}$ is the variance-covariance matrix of $\mathbf{S_p}$, $\mathbf{C_{BB}}$ the variance-covariance matrix of $\mathbf{S_B}$, $\mathbf{C_{pB}}$ and $\mathbf{C_{Bp}}$ are the variance-covariance matrices between $\mathbf{S_p}$ and $\mathbf{S_B}$ and vice versa. $\mathbf{m}$ is the design matrix of for fine-scale covariates of $\mathbf{S_p}$ .

Since the joint distribution is Gaussian the optimal predictions at unsampled point locations can be given as,

$$\mathbf{\hat{S}_p} = \mathbf{m}'\beta + \mathbf{C_{pB}}'\mathbf{C_{BB}}(\mathbf{S_B} - \mathbf{M}\beta) \tag{4}$$

where $\mathbf{\hat{S}_p}$ is the vector of predicted soil carbon at N point.

And the variance-covariance matrix for the prediction error is,

$$\mathrm{var}(\mathbf{S_p}|\mathbf{S_B}) = \mathbf{C_{PP}} - \mathbf{C_{pB}}\mathbf{C_{BB}}^{-1}\mathbf{C_{pB}}' \tag{5}$$

This shows that the inference of $\mathbf{S_p}$ from $\mathbf{S_B}$ is straightforward and resembles a simple kriging formulation (Cressie and Wikle, 2011). However, the estimation of point support variogram from the block support data is not a trivial task. Usually, the estimation is done via deregulation or deconvolution of the empirical variogram. Goovaerts

(2008); Gotway and Young (2007); Pardo-Igúzquiza and Atkinson (2007) proposed an iterative numerical procedure for the deconvolution of the block support variogram for irregular block support. Wang et al. (2015) suggested an integration approach, while Nagle et al. (2011) used a maximum likelihood approach. Gelfand et al. (2001) used Markov Chain Monte Carlo (MCMC) integration to estimate point support variogram from block support data where the parameters of point support variogram were given as non-informative priors. Truong et al.

(2014) used a similar approach, but the parameters of the point support variogram were given as informative priors. In this paper, the variogram parameters were inferred directly from the data via Bayesian estimation.

### 2.1 Bayesian inference

The likelihood function of Gaussian process S can be given as $f(\mathbf{S})|\beta, \boldsymbol{\theta}) = \mathrm{N}\left(\mathbf{m}\beta, \mathbf{C}(\boldsymbol{\theta})\right)$. $\mathbf{C}$ is a function of the

140 Euclidean distance h, with the vector of parameters $\boldsymbol{\theta}$. Then the likelihood of predictive distribution at point locations can be given as,

$$f\left(\mathbf{S_p}|\mathbf{S_B}\right) = \int f\left(\mathbf{S_p}|\mathbf{S_B} ; \beta, \boldsymbol{\theta}\right) f(\beta, \boldsymbol{\theta} \mid \mathbf{S_B}) \; d\beta \; d\boldsymbol{\theta} \tag{6}$$

Where $\mathbf{S}_p| \mathbf{S_B}, \beta, \boldsymbol{\theta}$ is distributed as

$$\mathrm{N}\left(\mathbf{M}\beta + \mathbf{C}'_{\mathbf{BP}}(\boldsymbol{\theta}) \mathbf{C}^{-1}_{\mathbf{pp}}(\boldsymbol{\theta}) \left(\mathbf{S_p} - \mathbf{m}\beta\right)\right), \mathbf{C_{BB}}(\boldsymbol{\theta}) - \mathbf{C}'_{\mathbf{BP}}(\boldsymbol{\theta}) \mathbf{C}^{-1}_{\mathbf{pp}}(\boldsymbol{\theta}) \mathbf{C_{Bp}}(\boldsymbol{\theta})) \tag{7}$$

Each entry of the distribution can be estimated through a Monte Carlo integration. Then we can replace Eq. 3 with $\hat{f}\left((\mathbf{S_p}, \mathbf{S_B})' \mid \beta, \boldsymbol{\theta}\right)$, where 'hat' denotes a Monte Carlo integration. Then it is apparent that,

$$\hat{f}\left((\mathbf{S_p}, \mathbf{S_B})' \mid \beta, \boldsymbol{\theta}\right) = f\left((\mathbf{S_p}, \widehat{\mathbf{S_B}})'\right) \mid \beta, \boldsymbol{\theta} \tag{8}$$





To predict at $\hat{\mathbf{S}}_{\mathbf{p}}$ we require $f(\hat{\mathbf{S}}_{\mathbf{p}}, \mathbf{S_B})$, which is given by Eq. 3. Using $\hat{f}\left((\mathbf{S_p}, \mathbf{S_B})' \mid \beta, \theta\right)$, and using Eq. 8 we can obtain $f\left((\mathbf{S_p}, \widehat{\mathbf{S_B}})'\right) \mid \beta, \theta$ to sample $\hat{\mathbf{S}}_{\mathbf{p}}$.

Then given the priors $[\beta, \theta]$, the Bayesian model is specified and a Monte Carlo fitting procedure can be carried out to maximise following log likelihood function. A complete description of deducing the Bayesian area to point inference can be found in Gelfand et al. (2001).

$$L(\beta, \theta | \mathbf{S_B}) = \frac{n}{2}\log(2\pi) + \frac{1}{2}\log|\mathbf{C_{BB}}| - \frac{1}{2}(\mathbf{S_B} - \mathbf{M\beta})^\mathsf{T}\mathbf{C_{BB}}^{-1}(\mathbf{S_B} - \mathbf{M\beta}) \tag{9}$$

Where $\mathbf{C_{BB}}$ is block to block covariance and can be approximately calculated as

$$\mathbf{C_{BB(i,j)}} = \begin{cases} \frac{c_1}{K} + \left(\frac{2}{K(K-1)}\right)\sum_{k=1}^{K}\sum_{l>k}^{K}\left(c_1 + c_0 - \gamma_p(s_k - s_l)\right), & i \neq j \\ \frac{1}{K^2}\left(c_1 + c_0 - \gamma_p(s_k - s_l)\right), & i = j \end{cases} \tag{10}$$

Where $i, j$ index are block supported observations, $k, l$ index discretised points within a block and K is the number of discretisation points per block (Truong et al., 2014). $c_1, c_0$ are the partial sill and the nugget component of the semivariogram; $\gamma_p$.

### 3 Case study

#### 3.1 Study area and coarse scale soil carbon map

We illustrate the methodology by disaggregating a regional topsoil organic carbon map of the University of Sydney E. J. Holtsbaum Agricultural Research Station (Fig. 1). The landholding, also known as "Nowley", is located in the Spring Ridge district of the central/north west slopes region of NSW, Australia. Its area is approximately 2300ha and is managed as a mixed farming enterprise of cattle grazing throughout the year, and wheat, barley, and canola in winter, and sorghum and sunflower in the summer. Cropping is performed on soils derived from basalt parent materials. The average annual rainfall is 600mm.

The available map is a New South Wales (NSW) topsoil carbon map, which was generated using a local regression kriging technique (Somarathna et al., 2016). These maps are analogues to the nationally available soil carbon maps of Australia which are available in separate layers corresponding to the depth intervals definition in the GlobalSoilMap product specifications (Arrouays et al., 2014). The standard soil depth layers are 0-5cm, 5-15cm, 15-30cm, 30-60cm, and 60-100cm.

This NSW soil carbon map has grid spacing 100m x 100m, with block support of the same size. Hence, map spatial resolution and the support are equal, which in turn mean that the pixel values that make up the map represent a predicted mean for the area of each pixel. Commonly the pixel values of a digital soil map are on point support, with the pixel value representing the value at a point usually at the centre of each pixel (Malone et al., 2013). Therefore, block kriging was used convert the maps to point supported maps before the analysis.

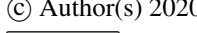



The topsoil of the study area was represented by the depth interval of 0-7.5 cm, as this was the depth interval used in previous work conducted at this site for soil carbon auditing (de Gruijter et al., 2016). The respective coarse (NSW) map for the 0-7.5cm depth interval was predicted by using mass-preserving splines (Malone et al., 2009) of the 0-5cm and 5-15cm standardised maps. This map will be hereafter referred to as the coarse scale map. It is this map that subsequent descriptions of spatial downscaling will refer to. The experimental variogram of the block supported coarse scale map is given in Fig. 2.

**3.2 Environmental covariates**

Elevation, topographic wetness index (TWI), Gamma radiometric Thorium, gamma radiometric potassium, Landsat band 4 and Landsat band 7 were used  as environmental covariates in the spatial downscaling model. Landsat data has a resolution of 30m. Since the coarse map was disaggregated into 10m resolution, the Landsat data were pan-sharpened using the panchromatic band (15m), and then resampled to 10m.

The elevation data had been collected during a ground survey using an all-terrain vehicle at with 20m swath. Then the data were processed using a local block kriging approach to produce an elevation data at a 10m resolution. Based on the DEM, the Topographic Wetness Index was derived. The gamma radiometric data was derived from aerial survey (Minty et al., 2009) which were resampled to 10m. A more detailed description of data collection and post-processing can be found in Malone et al. (2017).

**3.3 Model parameter inferencing using the MCMC simulation**

Suppose our observed soil carbon data (S) follows a second- order stationary Gaussian process which is characterised by the mean (trend ) and the variogram where $\mathbf{S_p} \sim N(\mathbf{S_p}, \gamma_p)$ where $\gamma_p = (h, \boldsymbol{\theta})$ ; is a function of Euclidian distance $h$ *and*  $\boldsymbol{\theta}$ vector of parameters of Matérn variogram (Eq. 14).  The block averages $\mathbf{S_B}$ have the same spatial mean as $\mathbf{S_p}$ but a different spatial structure (Gotway and Young, 2002). The block support covariance $\mathbf{C_{BB}}$ is also a function of h with parameter vector of $\boldsymbol{\theta}$ but has a different support.

$$C_{ij} = c_0\delta_{ij} + c_1\left[\frac{1}{2^{\upsilon-1}\Gamma(\upsilon)}\left(\frac{h}{r}\right)^{\upsilon}K_{\upsilon}\left(\frac{h}{r}\right)\right] \qquad (11)$$

where $C_{ij}$ is the covariance between observation i and j, h represents the separation distance between i and j, $\delta_{ij}$ denotes the Kronecker delta ($\delta_{ij}$ =1 if i=j and $\delta_{ij}$ =0 when I ≠ j), $c_0 + c_1$ signifies the sill variance, $K_{\upsilon}$ is the modified Bessel function of the second kind of order $\upsilon$. $\Gamma$ is the gamma function, r denotes the distance or 'range' parameter and $\upsilon$ is the spatial 'smoothness'. Calculating $\mathbf{C_{BB}}$ requires discretisation and it was done using a regular grid pattern, with equal 10m spacing between discretising points within a block.  Then $\mathbf{C_{BB}}$ was calculated according to Eq. 11.

The priors were given as informative priors and considered as normally distributed. The coarse scale soil carbon map was regressed against the selected covariates to estimate the approximate prior values for the coefficient of the linear trend (β). Gaussian priors were assigned for both spatial trend (β) and covariance

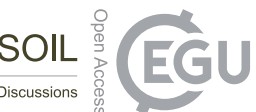

component (**θ**) of the spatial model. MCMC integration was carried out using the Metropolis Hasting algorithm. Metropolis Hasting is a computationally efficient algorithm that allows sampling from conditional distributions

(Diggle et al., 1998). Calibration of the model was done using two parallel chains of MCMC with 100 000 iterations.

### 3.4 Spatial prediction

The mean, 95% upper and lower CI limits of spatial model parameters were calculated using the values of the last 10 000 MCMC iterations. These calibrated parameters were used to predict SOC content at the fine scale. The predictions were made on to the 10m x 10m points based on Eq. 4.

### 3.5 Disaggregation using dissever

The proposed Bayesian model also compared with dissever regression based disaggregation approach. dissever is a regression-based downscaling approached proposed by Malone et al. (2012). dissever operates in two steps; initialisation and iteration. At initialisation step, the coarse map is resampled using nearest neighbour resampling technique to match the resolution of available fine scale covariates. Then the resampled map is regressed against the fine scale covariates. Next, the regressed fine map is upscaled through averaging, and then the mass balance

deviation (deviation factor; DF) from the coarse map is calculated. The regressed fine map is corrected using the DF, and then the algorithm progresses to the iteration step. In the iteration step, regression and upscaling continue until the mass balance deviation between the upscaled map and the original map is less than a set minimum (threshold). The following flow diagram (Fig. 3) is an illustrative description of the dissever algorithm. This differs from Bayesian ATPRK since it is not focused to address the change of support problem and it is a fine

gridding exercise. A detailed and comprehensive account of dissever can be found in Malone et al. (2012) and Malone et al. (2017).


Following the procedure mentioned above, the coarse scale carbon map of the area was disaggregated into 10m using dissever. The environmental covariates used in the Bayesian approach were similarly made available for dissever.






### 3.6 Independent validation

#### 3.6.1 Validation data

The accuracy of the model was tested using an independent dataset which consists of 127 soil samples from 0 - 7.5cm soil cores (cross-section area of 41 cm2 and height of 7.5 cm). The samples are considered to have point support. This data was collected over two separate soil surveys during 2014 and 2015 using stratified random sampling. Stratification was based on the SOC prediction fields generated from digital soil mapping. A detailed description about the stratified random sampling process can be found in de Gruijter et al. (2016). The carbon content of collected soil cores was measured using dry combustion by a vario MAX CNS analyser. The measured 265 carbon contents are in carbon concentrations (g of C per 100 g of soil).

#### 3.6.2 Validation statistics

The model accuracy was compared using Mean squared error (MSE) and Lins' concordance correlation coefficient (CCC) (Lin, 1989) between the disaggregated values and observed independent values at sampling 270 points.

CCC is given by

$$CCC = \frac{2\rho\sigma_x\sigma_y}{\sigma_x^2 + \sigma_y^2 + (\mu_x - \mu_y)^2}$$ (12)

Where $\rho$ is the correlation coefficient, $\sigma_x, \sigma_y$ are the variances of observed (x) and predicted (y) values and $\mu_x, \mu_y$
are the respective means. CCC is scaled between -1 and 1, the latter implying perfect agreement and the former implying perfect reverse agreement.

MSE is given by

$$MSE = \frac{\sum_{i=1}^{n'}(S_p - \hat{S}_{p'})^2}{n'}$$ (13)

where $\mathbf{S_p}$ is the observed values of SOC at independent sampling points, $\mathbf{\hat{S}_{p'}}$ predicted SOC vale at the fine scale and $n'$ number of independent SOC samples at a fine scale. MSE is an indication of the accuracy of predictions, while CCC indicates both accuracy and precision of the disaggregation exercise.

### 4 Results and Discussion

The use of LMM model as the spatial prediction model allows integration of fine-scale covariate information and spatial correlation in a single model. Unlike regression kriging in which regression and residual kriging are used in two phases of modelling, LMM is a one-step process. We used Bayesian inference techniques to estimate parameters of LMM to deliver uncertainty estimates of model parameters. This section provides a description of posterior parameters of the spatial model followed by a comparison of spatial disaggregated maps using Bayesian 290 ATPRK and Dissever.

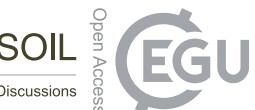

### 4.1 Bayesian parameter estimation of the spatial model

Fig. 4 illustrates the posterior statistics of the linear trend parameters. The upper diagonal panel displays correlation coefficients among the covariates, the lower diagonal depicts the scatter plots, while the diagonal shows the posterior distributions. The diagonal histograms are well defined which indicates that all parameters of the linear trend are relatively well-defined. The correlations among covariates are minimal, which indicate that there is no multicollinearity between the covariates.

Fig. 5 displays the posterior distribution of Matérn variogram parameters and their correlations. Posterior values of the range parameter are more-or-less similar to the prior values. However, the partial sill and nugget parameters are different from their priors. The posterior partial sill is higher than the prior sill, which agrees with the fact that the coarse scale variability of soil carbon is lower compared to the finer scale (Fig. 5b).

Table 1, 2 provides the estimates of the parameters of the spatial model along with the 95% confidence limits which is an indication of the uncertainty of the parameters. The estimated posterior parameters have narrow CI limits indicating the estimated mean provides a good approximation of the soil carbon content of the study area.

### 4.2 Spatial Disaggregation using Bayesian ATPRK

Fig. 6 displays the Bayesian ATPRK disaggregated soil carbon content of Nowley at 10m along with the 95% confidence interval limits. Unlike other methods which only give a single estimate of mean soil carbon content, the Bayesian technique provides a range/interval of the most probable value with a 95% confidence. These estimates are more reliable as they are generated using numerous repeated samplings. Accuracy and confidence in estimates are important parameters for planning, designing, and implementation of soil management. Therefore, Bayesian ATPRK provides a comprehensive output of soil carbon estimates at the farm scale.

### 4.3 Comparison of Bayesian ATPRK and Dissever disaggregation methods

The disaggregated maps from both techniques were upscaled using a spatial overlay to assess the accuracy of disaggregation. CCC values were calculated for both techniques. Fig. 7 shows the agreement between the cell values of coarse map and the upscaled disaggregated maps.




Concordance correlation between original coarse values and upscaled product were 87% and 96% for Bayesian ATPRK and dissever, respectively. dissever iterates until the agreement between upscaled and original coarse cell reaches a set minimum, while Bayesian ATPRK output is a result of one simulation considering both the correlation with the covariates and covariance structure of the residuals. This explains the stronger concordance associated with dissever.

Fig.8 illustrates the comparison between dissever and Bayesian ATPRK disaggregated product with the coarse scale source map. A closer examination reveals that Bayesian ATPRK product resembles the spatial pattern of SOC more closely than the dissever product while dissever seemingly over predicts the SOC content. Conversely, dissever is computationally more efficient compared to Bayesian ATPRK. Calculating the block to point and block to block distance matrixes is a computational expensive procedure. The whole study area cannot be computed in a single run using a desktop computer. Therefore, the study area has to be split into tiles, and

predictions were carried out for each of the tiles. This resulted in abrupt changes between two tiles in the Bayesian ATPRK disaggregated map. The use of parallel computing, high performing computing can improve the processing time.


### 4.4 Validation of Disaggregated maps

Independent validation results between disaggregated maps using dissever and Bayesian ATPRK indicate that Bayesian ATPRK technique is slightly more accurate compared to dissever produced output in general.

CCC value of Bayesian ATPRK was 23% while that of dissever was 18%. Greater CCC indicates more accurate and precise predictions. Mean squared error (MSE) of Bayesian ATPRK product was 0.45 while MSE for dissever was 0.49, which indicates that Bayesian ATPRK disaggregation is slightly more accurate than dissever disaggregation.

Although the validation CCC is low, it still is a sufficient approximation in terms of SOC as the SOC

prediction accuracy generally remains low. The coarse map is a product of regression for the whole region, which also has a lower concordance correlation with the sampled data. Through the disaggregation process, the errors tend to propagate widening the gap between the actual and predicted SOC content.

### 5 Conclusions

The study demonstrates that Bayesian techniques can be effectively used to address the change of support problem. Monte Carlo integration provides the estimates of point supported covariance structure using block supported data along with uncertainty estimates of parameter inference. This study used informative priors of covariance parameters as the soil carbon data were available for the study site. In the absence of prior information, priors can be given as non-informative priors.



Bayesian ATPRK provides an effective way of disaggregating coarse scale SOC maps. The output of this study facilitates policy making and soil management activities by providing the maps with confidence limits which lead to wider understanding of the SOC content of the area. However, the method is computationally expensive and the use of high performing computing techniques is recommended for faster and smoother predictions.

Further, the noise of coarse scale observation can be filtered through incorporating the noise variance in the diagonal of $\mathbf{C}_{BB}$. The use of point data (if available) in the spatial model can improve the accuracy of the disaggregation. Finally, the use of both of these techniques is recommended for lower uncertainty and improved accuracy of the disaggregation process.


**Acknowledgements**

The authors are grateful to the Australian Dep. of Agriculture, Round 2, Filling the Research Gap Program for supporting this research (Grant No. 1194105-66). The authors extend thank Gerard Heuvelink, Wageningen University, for sharing his R code on area to point kriging and providing constructive insight to this paper.

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

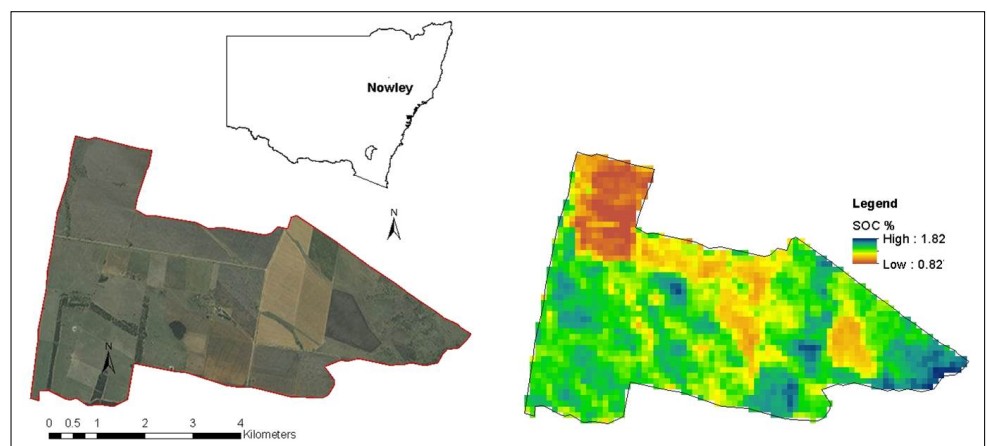

Figure1: The study area (left) and the coarse scale topsoil soil organic carbon map (right)




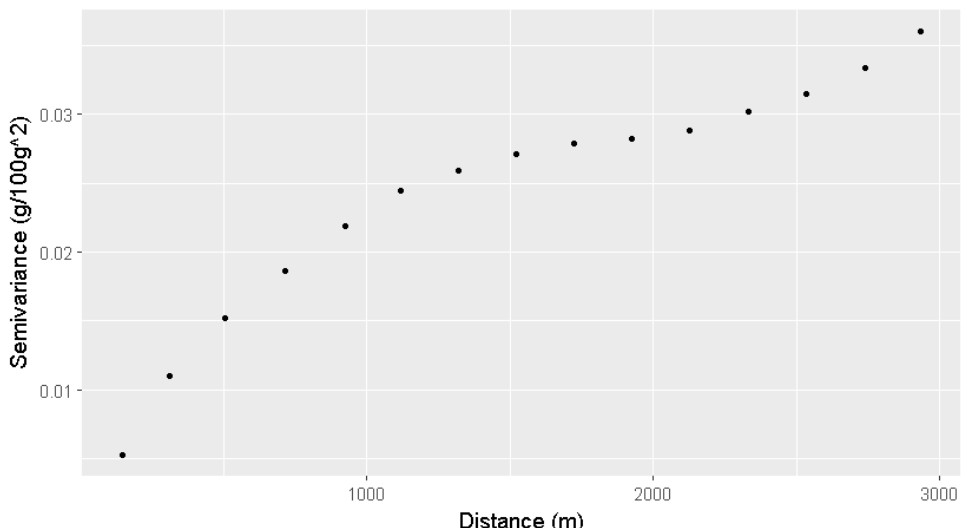

545 Figure 2: Block support experimental variogram of the coarse scale (100 m resolution) soil carbon map

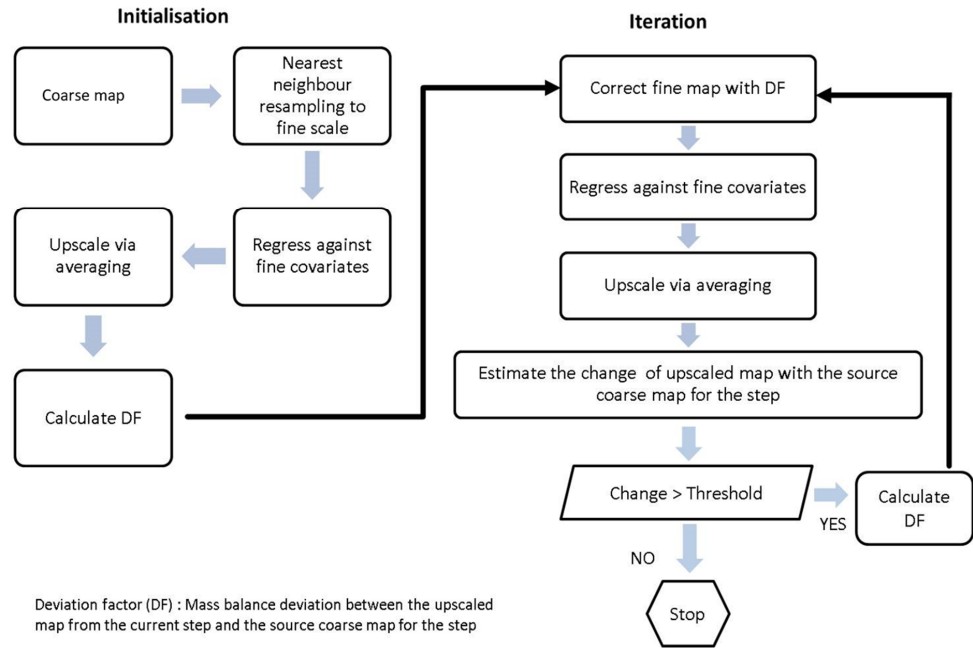

550 Figure 3: Schematic representation of dissever disaggregation technique




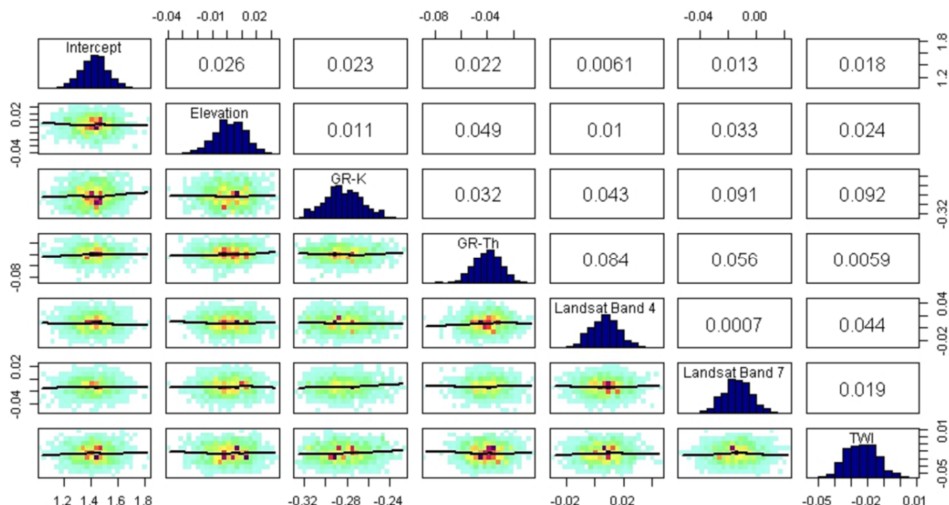

Figure 4: Correlations and posterior distribution of linear trends parameters of the spatial model

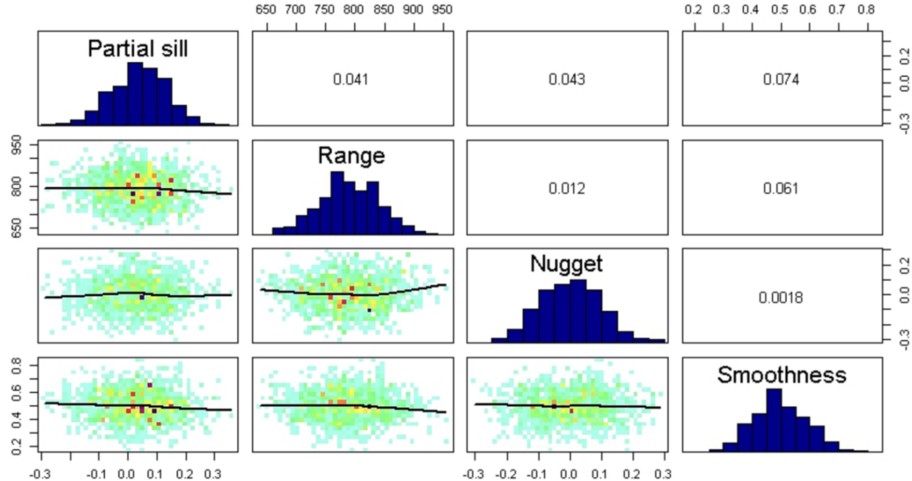


a.





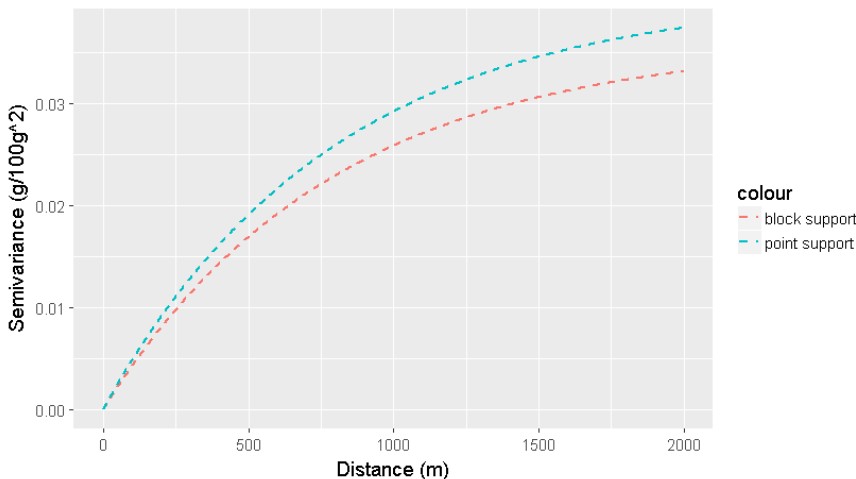

b

Figure 5: (a) Correlations and posterior distribution of variogram parameters (b) and the point and block support
variograms







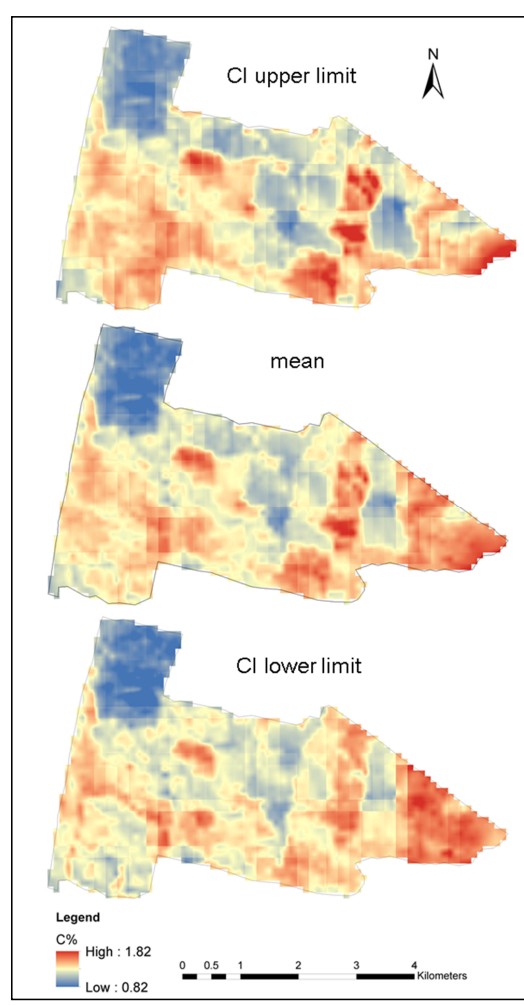

Figure 6: Bayesian ATPRK disaggregated map with the 95% confidence limits






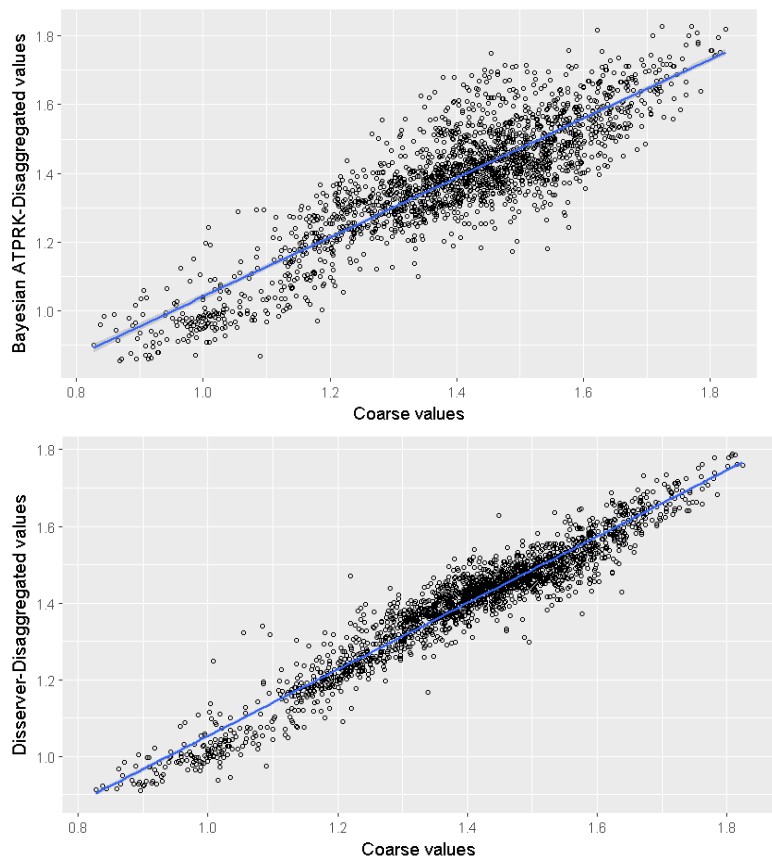

Figure 7: Correlation plots between upscaled disaggregated values from Bayesian ATPRK, dissever and original
coarse scale block values.



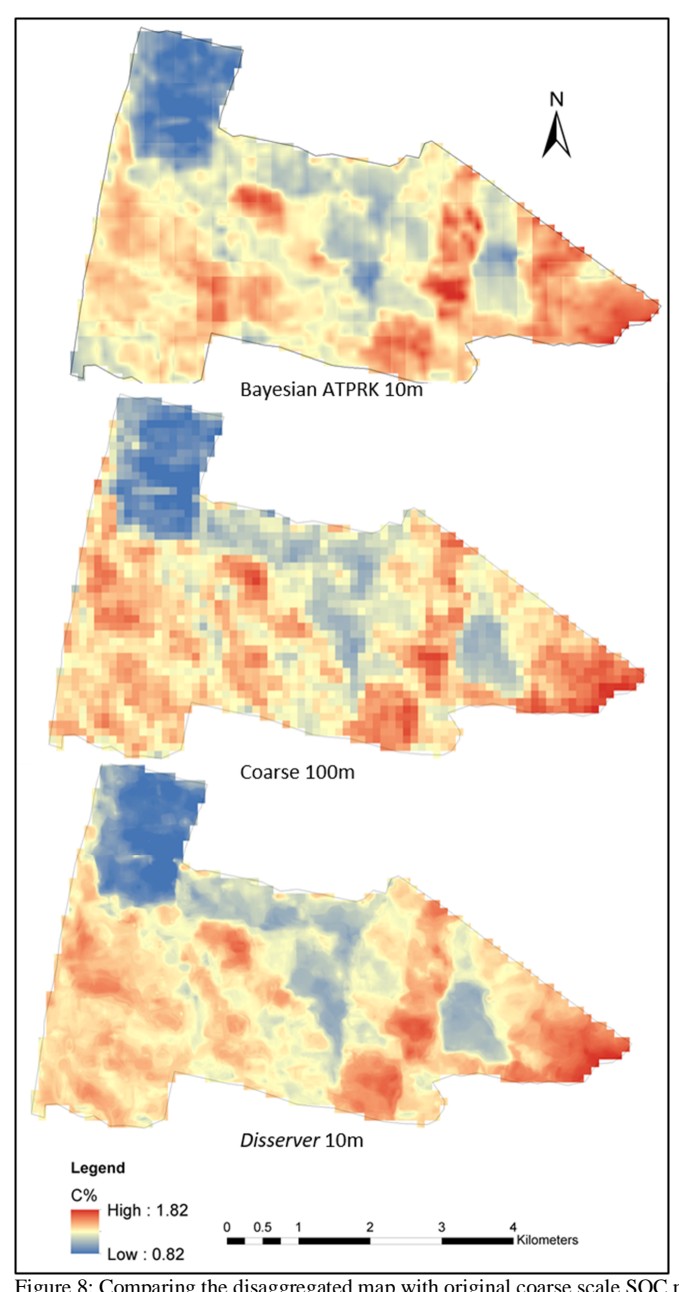

Figure 8: Comparing the disaggregated map with original coarse scale SOC map






Table 1. Posterior parameters of the linear trend model.

|  | $\beta_0$ | $\beta_1$ | $\beta_2$ | $\beta_3$ | $\beta_4$ | $\beta_5$ | $\beta_6$ |
|---|---|---|---|---|---|---|---|
| Lower CI limit | 1.360 | -0.003 | -0.291 | -0.045 | 0.002 | -0.019 | -0.030 |
| Mean | 1.416 | 0.002 | -0.281 | -0.040 | 0.007 | -0.014 | -0.024 |
| Upper CI limit | 1.471 | 0.007 | -0.270 | -0.034 | 0.012 | -0.009 | -0.018 |


$\beta_0$, $\beta_1$, $\beta_2$, $\beta_3$, $\beta_4$, $\beta_5$ and $\beta_6$ are the coefficients of the linear trend for intercept, elevation, TWI, gamma radiometric thorium, gamma radiometric potassium, Landsat band 4 and Landsat band 7 parameters respectively.

Table 2. Posterior parameters of the point support Matern variogram model.

| Method | Nugget | sill | range | smoothness |
|---|---|---|---|---|
| Lower CI limit | -0.05 | -0.02 | 759.10 | 0.45 |
| Mean | 0.005 | 0.034 | 788.24 | 0.50 |
| Upper CI limit | 0.06 | 0.09 | 817.37 | 0.57 |
