# Peer review of "Disaggregating a Regional Extent Digital Soil Map using Bayesian Area-to-Point Regression Kriging for Farm-Scale Soil Carbon Assessment"

_SOIL, 2019_

## Referee Comment (RC1) · Anonymous Referee #1 · 12 Mar 2020

General comments: This manuscript proposed the use of Bayesian Area-to-Point Regression Kriging (ATPRK) for spatial disaggregation of regional scale digital soil map to farm scale. This method is able to provide the uncertainty estimate associated with the disaggregation process. Throughout it had lower concordance correlation with the coarse map than dissever algorithm, the independent data showed that Bayesian ATPRK had a higher concordance correlation than dissever. Generally, the proposed method is interesting as it can provide the uncertainty related to the disaggregation process. However, from my point of view, the major uncertainty of disaggregated map

comes from the original map, thus it is more important to incorporate the uncertainty of the original map into disaggregation. Is the proposed method able to integrate this? Another limitation is the Results and Discussion section, where authors focused mostly on the results and rarely provided a more general discussion. I look forward to seeing the feedback from the authors on these two issues.

Specific comments:

Line 21: Why only mentioned underestimation? How about overestimation? Line 35: What is the difference in computing time? I know Bayesian ATPRK is a one-step process, and it saves computing time? Because sometimes an iteration process can also be efficient. Line 45: It is not "five standard soil depths" but "six standard depth intervals". Lines 71-72: I do not agree with the statement here, DSMART can also produce maps in a fine grid. Line 123: What do you mean "at N point"? Line 136: Since you listed many methods for deconvolution of the empirical variogram, it is better to show here why you chose Bayesian estimation. Lines 187-189: Do you think the proposed framework can integrate the uncertainty from the fitted mass-preserving splines as well as from the uncertainty estimates associated to these two SOC maps (0-5, 5-15 cm)? I understand that the Bayesian ATPRK can provide the uncertainty in the step of disaggregation, however, I think the major uncertainty comes from the original map, which should be not ignored in disaggregation. Lines 196-197: Why you choose 10 m as the final resolution? The finest covariate used here is in10 m (Elevation data), I think you should mention it in the very beginning. Line 268: I suggest adding PICP to assess the uncertainty. Lines 341-342: reduce the processing time? However, parallel computing can still not solve abrupt changes between two tiles. Maybe you need a laptop with larger RAM? Line 579: Figure 6 still showed an obvious block effect. In order to compare the original SOC map (Figure 1), please use the same legend and then we can tell the difference between them. Line 595: The map from Disserver looks smoother.

Technical corrections Line 21: DSM has not been defined in the previous texts. Line 48: DSM has been defined before. Line 91: SOC has not been defined yet. Line 200:

DEM has not been defined yet and Topographic Wetness Index can be replaced by TWI here. Line 209: and should not be in italic. Please have a careful check of all the parameters which should be in italic. Line 260: In "cm2", 2 should be in superscript. Line 280: predicted SOC values. Line 285: LMM has not been defined. Line 290: In other texts, dissever is used. Please make it consistent. Line 539: Please use either Figure or Fig. in the text. Currently, it is a mix. Line 565: Better to put (a) and (b) in the upper right position.

---

## Author Comment (AC1) · 20 Mar 2020

First of all, the authors would like to thank you providing constructive comments on this manuscript. We have identified the uncertainty of the original should also be considered in the disaggregation process. and mentioned this in our conclusions section as a recommendation. Our main focus of this paper was to estimate the uncertainty of the disaggregation process which is overly neglected in the current literature. However, the proposed Bayesian method is capable of handling the uncertainty of the original map by incorporating the uncertainty as a noise variance in the covariance matrix; CBB.

The noise variance can be filtered out by including in the diagonal of the covariance matrix.

Also, we agree with the reviewer that the discussion section should be little more generalised. We will revise the paper accordingly.

---

## Referee Comment (RC2) · Anonymous Referee #2 · 20 May 2020

The manuscript is well-written and follows up from earlier work of Malone et al. 2017 on disaggregation of coarse scale SOC maps. This is particular important for facilitating management at the field/farm scale. Although I am not a geostatistical expert, the data anlysis is sound and the results are convincing, although not much different from the ones obtained by the 'dissever' approach. Lines 19-20 Is not there a verb missing in the sentence starting with 'Disaggregating' ? Line 21 Define 'DSM' at first time use. Line 52 I appreciate that you try to use the original meaning of scale 'large' and 'small', but the word 'or' might create confusion. How about replacing 'or' by 'i.e.'? Line 53 ..into

two categories... Line 62 Please rephrase this sentence and break it up. e.g. They demonstrated that a Gaussian process model can be used as a novel image fusion technique. They used images from unmanned aerial vehicle flown over agricultural fields for panchromatic sharpening of low-resolution satellite images.

Line 65 Avoid using 'approach' twice in one sentence. Please rephrase line 74 are not there some commas missing? Line 79 ..and also ancillary covariates such as the 'scorpan factors' (McBratney et al., 2003; Kerry et al., 2012). Line 83 ...underlying.. Line 89 ..soil organic carbon (SOC° maps.. SOC is arguably.... Line 236 ...approach... Line 264 Specify the manufacturer and the country for the Variomax. Did you check the samples for inorganic carbon (carbonates)? Line 285 Has 'LMM' been defined before? Avoid using 'model' twice in one sentence. Line 351 The MSE has units.

---

## Author Comment (AC2) · 29 May 2020

We would like to thank the reviewer for his/her thoughtful comments and efforts towards improving our manuscript. We will revise the manuscript accordingly.

---

## Author Response (AR1)

[revised manuscript text omitted]

To predict at $\hat{S}_p$ we require $f(\hat{S}_p, S_B)$, which is given by Eq. 3. Using $\hat{f}\left((S_p, S_B)' \mid \beta, \theta\right)$, and using Eq. 8 we can obtain $f\left((S_p, \widehat{S_B})'\right) \mid \beta, \theta$ to sample $\hat{S}_p$.

Then given the priors $[\beta, \theta]$, the Bayesian model is specified, and a Monte Carlo fitting procedure can be carried out to maximise following log likelihood function. A complete description of deducing the Bayesian area to point inference can be found in Gelfand et al. (2001).

$$L(\beta, \theta \mid S_B) = \frac{n}{2}\log(2\pi) + \frac{1}{2}\log|C_{BB}| - \frac{1}{2}(S_B - M\beta)^T C_{BB}^{-1}(S_B - M\beta) \tag{9}$$

[revised manuscript text omitted]

a.

[Figure]

610

615

b

[Figure]

Figure 5: (a) Correlations and posterior distribution of variogram parameters (b) and the point and block support variograms.

620

625

[Figure]

630    Figure 6: Bayesian ATPRK disaggregated map with the 95% confidence limits.

635

[Figure]

640

Figure 7: Correlation plots between upscaled disaggregated values from Bayesian ATPRK, dissever and original coarse scale block values.

[Figure]

Figure 8: Comparing the disaggregated map with original coarse scale SOC map.

655

Table 1. Posterior parameters of the linear trend model.

|  | $\beta_0$ | $\beta_1$ | $\beta_2$ | $\beta_3$ | $\beta_4$ | $\beta_5$ | $\beta_6$ |
|---|---|---|---|---|---|---|---|
| Lower CI limit | 1.360 | -0.003 | -0.291 | -0.045 | 0.002 | -0.019 | -0.030 |
| Mean | 1.416 | 0.002 | -0.281 | -0.040 | 0.007 | -0.014 | -0.024 |
| Upper CI limit | 1.471 | 0.007 | -0.270 | -0.034 | 0.012 | -0.009 | -0.018 |

$\beta_0$, $\beta_1$, $\beta_2$, $\beta_3$, $\beta_4$, $\beta_5$ $and \beta_6$ are the coefficients of the linear trend for intercept, elevation, TWI, gamma radiometric thorium, gamma radiometric potassium, Landsat band 4 and Landsat band 7 parameters respectively.

660

Table 2. Posterior parameters of the point support Matern variogram model.

| Method | Nugget | sill | range | smoothness |
|---|---|---|---|---|
| Lower CI limit | -0.05 | -0.02 | 759.10 | 0.45 |
| Mean | 0.005 | 0.034 | 788.24 | 0.50 |
| Upper CI limit | 0.06 | 0.09 | 817.37 | 0.57 |

665

670

675

680

685

690

695

**Response to Reviewer's comments**

We would like to thank the reviewers for providing constructive comments and efforts towards improving this manuscript. We highlight the reviewers' comments in red and our responses are given in black.

**Reviewer 1 comments and Responses**

General comments: This manuscript proposed the use of Bayesian Area-to-Point Regression Kriging (ATPRK) for spatial disaggregation of regional scale digital soil map to farm scale. This method is able to provide the uncertainty estimate associated with the disaggregation process. Throughout it had lower concordance correlation with the coarse map than dissever algorithm, the independent data showed that Bayesian ATPRK had a higher concordance correlation than dissever. Generally, the proposed method is interesting as it can provide the uncertainty related to the disaggregation process. However, from my point of view, the major uncertainty of disaggregated map comes from the original map, thus it is more important to incorporate the uncertainty of the original map into disaggregation. Is the proposed method able to integrate this? Another limitation is the Results and Discussion section, where authors focused mostly on the results and rarely provided a more general discussion. I look forward to seeing the feedback from the authors on these two issues

We have identified the uncertainty of the original should also be considered in the disaggregation process. We have mentioned this in our conclusions section as a recommendation. Our main focus of this paper was to estimate the uncertainty of the disaggregation process which is overly neglected in the current literature. However, the proposed Bayesian method is capable of handling the uncertainty of the original map by incorporating the uncertainty as a noise variance in the covariance matrix. $C_{BB}$. This enables filtering the noise. Also, we have added more detailed discussion about filtering measurement errors. (lines 341 -348)

Another limitation is the Results and Discussion section, where authors focused mostly on the results and rarely provided a more general discussion.

We agree with the reviewer that the discussion section should be little more generalised. Therefore, a more generalised discussion was added at the of the results and discussion section (lines 341 -361).

Specific comments: Line 21: Why only mentioned underestimation? How about overestimation?

Here we are referring to the fact that in DSM literature the uncertainty estimates of the disaggregation process are often neglected.

Line 35: What is the difference in computing time? I know Bayesian ATPRK is a one-step process, and it saves computing time? Because sometimes an iteration process can also be efficient.

The difference in computing time is approximately 3 hrs

Line 45: It is not "five standard soil depths" but "six standard depth intervals".

The typo was corrected

Lines 71-72: I do not agree with the statement here, DSMART can also produce maps in a fine grid.

Here, we are referring to the fine riding process where we disaggregated a coarse grid cell (100m) into many (10 x 10) fine grid cells. While the rasters created by DSMART based on the most probable class assigned after generating realisations from classification trees. The classification trees are built using the values taken from random samples from each soil class polygon from the legacy maps.

Line 123: What do you mean "at N point"? Line 136: Since you listed many methods for deconvolution of the empirical variogram, it is better to show here why you chose Bayesian estimation.

We have included the objective of choosing Bayesian techniques to infer the point support variogram. (line 137-138)

Lines 187-189: Do you think the proposed framework can integrate the uncertainty from the fitted mass-preserving splines as well as from the uncertainty estimates associated to these two SOC maps (0-5, 5-15 cm)?

Yes, the uncertainty from all sources can be accounted using the proposed Bayesian techniques. This is also included in the added new section to the results and discussion section.

I understand that the Bayesian ATPRK can provide the uncertainty in the step of disaggregation, however, I think the major uncertainty comes from the original map, which should be not ignored in disaggregation.

We have identified this as a limitation of the study, and it was further elaborated in this revised version (line 341-361.)

Lines 196-197: Why you choose 10 m as the final resolution? The finest covariate used here is in10 m (Elevation data), I think you should mention it in the very beginning.

There were two reasons behind choosing 10m as the fine resolution. One is the resolution of available data at fine

scale and secondly, we believe that 10m maps are informative enough for supporting decision making process at fine scale. In anotherword,10m maps would capture most of the spatial variability of SOC and therefore we selected 10m as our fine scale's resolution.

Line 268: I suggest adding PICP to assess the uncertainty.

We present CI estimates of model parameters and maps. CI values provide an insight to the uncertainty estimates.

Therefore, we decided not to include PICPs

Lines 341-342: reduce the processing time? However, parallel computing can still not solve abrupt changes between two tiles. Maybe you need a laptop with larger RAM?

Yes, it needs a bigger RAM to disaggregate the whole study area at once.

Line 579: Figure 6 still showed an obvious block effect. In order to compare the original SOC map (Figure 1),

please use the same legend and then we can tell the difference between them.

Figure 1 was changed to the same legend.

Line 595: The map from Disserver looks smoother.

Disserver disaggregate the whole study area at once resulting a smoother map. Therefore, the maps are smoother.

Technical corrections Line 21: DSM has not been defined in the previous texts. Line 48: DSM has been defined

before.

Respective corrections were made.

Line 91: SOC has not been defined yet.

SOC defined

Line 200: C2 SOILD Interactive comment Printer-friendly version Discussion paper DEM has not been defined

yet and Topographic Wetness Index can be replaced by TWI here.

DEM was defined and the Topographic Wetness Index was replaced by TWI

Line 209: and should not be in italic. Please have a careful check of all the parameters which should be in italic.

Line 260: In "cm2", 2 should be in superscript.

Respective corrections were made

780 Line 280: predicted SOC values.

Typo was corrected

Line 285: LMM has not been defined.

LMM was defined

Line 290: In other texts, dissever is used. Please make it consistent.

785 Line 539: Please use either Figure or Fig. in the text. Currently, it is a mix.

The term "Figure" has been used as the title of figures while "Fig." was used in the text when the figures are described

Line 565: Better to put (a) and (b) in the upper right position.

Changed the positions of a and b

790

**Reviewer 2 comments and responses**

The manuscript is well-written and follows up from earlier work of Malone et al. 2017 on disaggregation of coarse scale SOC maps. This is particular important for facilitating management at the field/farm scale. Although I am not a geostatistical expert, the data anlysis is sound and the results are convincing, although not much different

795 from the ones obtained by the 'dissever' approach.

Lines 19-20 Is not there a verb missing in the sentence starting with 'Disaggregating' ?

The typo was corrected

Line 21 Define 'DSM' at first time use.

DSM was defined

800 Line 52 I appreciate that you try to use the original meaning of scale 'large' and 'small', but the word 'or' might create confusion. How about replacing 'or' by 'i.e.'?

Line 53 ..into two categories. . .

Typo was corrected

Line 62 Please rephrase this sentence and break it up. e.g. They demonstrated that a Gaussian process model can

805 be used as a novel image fusion technique. They used images from unmanned aerial vehicle flown over agricultural fields for panchromatic sharpening of low-resolution satellite images.

The sentence was rephrased.

All the following typos were corrected.

Line 65 Avoid using 'approach' twice in one sentence.

810 Please rephrase line 74 are not there some commas missing?

Line 79 ..and also ancillary covariates such as the 'scorpan factors' (McBratney et al., 2003; Kerry et al., 2012).

Line 83 . . .underlying..

Line 89 ..soil organic carbon (SOC∘ maps.. SOC is arguably. . ..

Line 236 . . .approach. . .

815 Line 264 Specify the manufacturer and the country for the Variomax. Did you check the samples for inorganic carbon (carbonates)?

Line 285 Has 'LMM' been defined before? Avoid using 'model' twice in one sentence.

Line 351 The MSE has units.